# Reducing Videoconferencing Fatigue through Facial Emotion Recognition

**Jannik Rößler** [1], **Jiachen Sun** [2] **and Peter Gloor** [3],*

1    Cologne Institute for Information Systems, University of Cologne, 50923 Cologne, Germany; roessler@wim.uni-koeln.de
2    School of Electronics and Information Technology, Sun Yat-sen University, Guangzhou 510006, China; sunjch6@mail2.sysu.edu.cn
3    MIT Center for Collective Intelligence, Massachusetts Institute of Technology, Cambridge, MA 02142, USA
*    Correspondence: pgloor@mit.edu

**Abstract:** In the last 14 months, COVID-19 made face-to-face meetings impossible and this has led to rapid growth in videoconferencing. As highly social creatures, humans strive for direct interpersonal interaction, which means that in most of these video meetings the webcam is switched on and people are "looking each other in the eyes". However, it is far from clear what the psychological consequences of this shift to virtual face-to-face communication are and if there are methods to alleviate "videoconferencing fatigue". We have studied the influence of emotions of meeting participants on the perceived outcome of video meetings. Our experimental setting consisted of 35 participants collaborating in eight teams over Zoom in a one semester course on Collaborative Innovation Networks in bi-weekly video meetings, where each team presented its progress. Emotion was tracked through Zoom face video snapshots using facial emotion recognition that recognized six emotions (happy, sad, fear, anger, neutral, and surprise). Our dependent variable was a score given after each presentation by all participants except the presenter. We found that the happier the speaker is, the happier and less neutral the audience is. More importantly, we found that the presentations that triggered wide swings in "fear" and "joy" among the participants are correlated with a higher rating. Our findings provide valuable input for online video presenters on how to conduct better and less tiring meetings; this will lead to a decrease in "videoconferencing fatigue".

**Keywords:** facial emotion recognition; social network analysis; video meetings

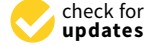



## 1. Introduction

Famous speeches such as Martin Luther King's "I have a dream" in 1963 or Barack Obama's election victory speech in 2008 are known for their rhetorical manifestations, the selection of inspiring words and the use of vivid and metaphorical language. However, another important driver for a great speech or presentation is the method by which presenters express their information by using emotions [1]. Presentations which make use of emotional experiences and trigger an emotional response from the listeners are more likely to gain the attention of the audience than compared to emotionless presentations [2]. For example, consumers primarily use emotions rather than information when evaluating brands [3]. Moreover, politicians often express anger and sadness to express empathy and worriedness about the topic at hand [1]. Furthermore, brain researchers have showed that humans have to be emotional in order to be memorable [4].

Emotions in presentations can be expressed with multiple methods: the choice of vocabulary, e.g., using emotional words such as "sad", "cry", and "loss" when expressing sadness; the method the presenter communicates, e.g., aggressively vs. sensitively; or the gestures and facial expressions the presenter makes with his/her body and face, e.g., looking happy, sad, angry, disgusted, or surprised [5]. All of the above conveys the presenter's feelings. In particular, facial emotion recognition is a broad and well-studied

research stream which has recently received a lot of attention due to the rising development of deep learning based methods. Thanks to deep neural networks with convolutional neural networks (CNN), recurrent neural networks (RNN), and long short-term memory (LSTM) models in particular, researchers can extract emotions from facial expressions with a high degree of accuracy. For example, a large number of researchers demonstrated that their deep learning based methods perform very well on various facial emotion recognition data sets (see [6] for an overview). In addition, Choi and Song [7] showed that their framework—a combination of CNN and LSTM—is not only robust and effective on artificial facial expression data sets (actors playing various emotions), but is also robust and effective on data sets collected in the wild.

Leveraging facial emotion recognition, some researchers analyzed the relationship between facial expressions and the learning process, especially in e-learning environments, to monitor and measure student engagement [8], to create personalized educational e-learning platforms [9], or to provide personalized feedback to improve the learning experience [10]. Our study joins this research stream by analyzing the relationship between speaker and listener's facial expressions and the quality of the presentations. Such an evaluation could help researchers and practitioners better understand the inherent relationship between emotions and presentations. Moreover, the insights could be used to create better presentations, which renders the presentation content more memorable and fosters better knowledge transfer. Hence, our research aims to answer the question of which sequence of emotions among the audience, expressed through their facial expressions, lead to good presentations and which do not. In particular, we hypothesize that certain combinations of emotions conveyed by the presenter or shown by the audience are indicators of higher perceived presentation quality.

The lack of such an analysis is likely due to the difficulty of analyzing facial expressions not only of the presenter but also of the audience. Furthermore, presentations need to be quantified objectively. The current Coronavirus pandemic presents a unique opportunity for such an analysis, as pupils, students, companies, and other organizations are shifting towards online platforms such as Zoom, Microsoft Teams, and Skype, which allow presentations to be held in front of a camera. This makes it easy to determine the presenter's and listener's emotions by using Deep Learning. Furthermore, such platforms often enable built-in functions which can be used to assess the quality of the presentation through the audience, for example, by using real-time questionnaires.

For this reason, we recorded and evaluated presentations from a virtual seminar called Collaborative Innovation Networks (COINs) over the course of twelve weeks with a total of seven two hour meetings using Zoom. There were eight student teams that consisted of three to five students. Each team presented its project progress in each of the seven two hour sessions spread out over the twelve weeks. In each team presentation, at least one or more team members presented their progress in ten minutes. After each presentation was held, the supervisor and the audience immediately evaluated it via a questionnaire. The facial recordings of the presentations were used to determine emotions of presenter, supervisor, and audience. We extracted the emotions using a deep neural network, a CNN based on the VGG16 architecture [11], which was trained prior to the recordings using self-labelled data and various facial emotion recognition data sets such as the Extended Cohn-Kanade (CK+) [12], the Japanese Female Facial Expressions (JAFFE) [13], and the BU-3DFE [14] data set.

Using the emotions from the supervisor and the audience as well as the presentations scores, we found that:

- The happier the speaker is, the happier and less neutral the audience is;
- The more neutral the speaker is, the less surprised the audience is;
- Triggering diverse emotions such as happiness, neutrality, and fear leads to a higher presentation score;
- Triggering too much neutrality among the participants leads to a lower presentation score.

The remainder of this paper is organized as follows. In Section 2, we review relevant literature with respect to facial emotion recognition and the relationship between emotions and the quality of presentations. In Section 3, we introduce the experimental setup and the research method. Results and discussions are presented in Sections 4–6, respectively. Finally, Section 7 summarizes the paper.

## 2. Related Work

We review the relevant literature with respect to, firstly, the application of deep learning in the context of facial emotion recognition, and secondly, the relationship between emotions and the quality of presentations.

### 2.1. Facial Emotion Recognition

Facial emotion recognition (FER) is a stream of work in which researchers in the area of computer vision, affective computing, human–computer interaction, and human behavior deal with the prediction of emotions using facial expressions in images or videos [15].

FER literature can be divided into two groups according to whether the features are handcrafted or automatically generated through the output of a deep neural network [6]. Furthermore, FER research can be distinguished according to whether the underlying emotional model is based on discrete emotional states [16] or on continuous dimensions, such as valence and arousal [17]. In the former, researchers share a consistent notion of emotions as discrete states, although different determinations exist with respect to what these basic emotions are. For example, Ekman and Oster [16] identified the five basic emotions as happiness, anger, disgust, sadness, and fear/surprise. Panksepp [18] defines play, panic, fear, rage, seeking, lust, and care as basic emotions. In the continuous dimension model, emotions are described by two or three dimensions containing valence or pleasant as one dimension and arousal or activation as the other dimension [19]. Considering that we leverage discrete emotional states in our work as well as the fact that relatively few studies develop and evaluate algorithms for the continuous dimension model [19], especially for the handcrafted feature generation process, we will only focus on the discrete emotional model type that distinguishes between handcrafted and deep learning based approaches.

FER approaches that use handcrafted features are usually deployed in three steps: face and facial component detection, feature extraction, and expression classification [6]. In the first step, faces and facial components (e.g., eyes and mouth) are detected in an input image. In the second step, various spatial and temporal features such as Histogram of Gradients (HoG), Local Binary Pattern (LBP), or Gabor Filters are extract from the facial components. Finally, machine learning algorithms such as Support Vector Machine (SVM) or Random Forests use the extracted features to recognize emotional states. Researchers have shown that manually extracting features can lead to accurate results [20–23] and practitioners use such approaches because, compared to deep learning based methods, manually extracting features requires much less computational resources [6].

However, due to the rise in the size and variety of data sets and thanks to recent developments in deep learning, deep neural networks have been the most appropriate technique in all computer vision tasks including FER [15]. Many researchers showed the superiority of deep learning algorithms, with CNNS, RNNS, and LSTM models in particular, over handcrafted approaches [24–33]. The main advantage of neural networks is their ability to enable "end-to-end" learning, whereby features are learned automatically from the input images [6]. One of the most deployed neural networks in the context of facial emotion recognition is the CNN [6], which is a special kind of neural network for processing images by using a convolutional operation [34]. The advantage of such a neural network is that it can take into account spatial information that is location-based information. However, CNNs "cannot reflect temporal variations in the facial components" [6] (p. 8) and thus, more recently, RNNs or LSTM models in particular, have been combined with CNNs to capture not only the spatial information but also the temporal features [7,26,30–33].

In our study, we use a deep learning based method, a CNN, to recognize a subset of the Ekman model's emotions which are the following: anger, fear, happiness, sadness, and surprise. Furthermore, we augmented the emotions with the neutral expression as a state of control for the recognition results on emotions, similar to Franzoni et al. [35]. Moreoever, as suggested by Kim et al. [36] and Kuo et al. [37], we trained our CNN on a merged data set that consisted of various original FER data sets such as CK+ [12], JAFFE [13], and BU-3DFE [14] to alleviate the problem of overfitting and to further improve its robustness. Finally, we leveraged a well-known pre-trained model architecture, VGG16 [11], to further improve the effectiveness of our CNN [38].

### 2.2. Presentations and Emotions

Deep neural networks have proved successful in a plethora of emotion recognition challenges such as facial emotion recognition [39], speech emotion recognition [40], or multimodal emotion recognition [41]. However, the use of such methods to predict emotions and the subsequent investigation into which extent emotions influence the quality of a presentation is limited, although some studies analyze the relationship between facial expressions and the learning process; this is especially seen in e-learning environments.

Zeng et al. [5] developed a prototype system which uses multimodal features including emotion information from facial expressions, text, and audio to explore emotion coherence in presentations. By analyzing 30 TED talk videos and examining two semi-structured expert interviews, the authors demonstrated that the proposed system can be used to, firstly, teach speakers to express emotions more effectively improving presentations and, secondly, teach presenters to include joke-telling to promote personalized learning. Although the authors also stress the importance of emotions in presentations, our study differs significantly from the work by Zeng et al. [5] in that we incorporate the emotions from the speaker and the audience as opposed to only analyzing the emotions from the speaker and, rather than considering expert knowledge, we use a score given after each presentation by all participants except the presenter to analyze the influence of emotions on the perceived outcome of video meetings. Finally, while Zeng et al. [5] developed a system which describes emotion coherence on different channels throughout a presentation, we investigate which emotional patterns lead to great presentations.

In their work, Chen et al. [42] use multimodal features including speech content, speech delivery (fluency, pronunciation, and prosody), and nonverbal behaviors (head, body, and hand motions) to automatically assess the quality of public speeches. The authors collected 56 presentations from 17 speakers whereby each speaker had to perform four different tasks. The presentations were scored by human raters on ten dimensions, such as vocal expression and paralinguistic cues, to engage the audience. The authors showed that multimodal features can be leveraged by machine learning algorithms, which is a random forest and support vector machine, to assess the performance of public speeches. Although the study took into account verbal and nonverbal behaviors, it neither considered emotions from speech nor from nonverbal behaviors such as facial expressions. Furthermore, the authors did not use sophisticated deep learning algorithms (e.g., convolutional neural networks) to predict emotions.

In another stream of research, researchers leveraged facial emotion recognition to improve educational e-learning platforms and thus the learning experience by providing personalized programs [9], personalized feedback [10], and by measuring student engagement [8]. For example, Carolis et al. [10] developed a tool for emotion recognition from facial expressions to analyze difficulties and problems of ten students during the learning process in a first year psychology course. The system was used to detect various emotions, such as enthusiasm, interest, concentration, and frustration during two situations: while presenting prerecorded video lectures to the students and while the students participated in an online chat with a teacher. The authors found that emotions can be an "indicator of the quality of the student's learning process" [10] (p. 102). For example, they argued that energy is a key factor to avoid boredom and frustration.

The study by Carolis et al. [8] is most similar to our work. The authors automatically measured the engagement of 19 students by analyzing facial expressions, head movements, and gaze behavior from 33 videos which contained more than five and a half hours of recordings. The collected data were related with a subjective evaluation of the engagement coming from a questionnaire with four dimensions: challenge, skill, engagement, and perceived learning. The authors found that the less stressed and more relaxed students are, the more engaged they appeared to be. Furthermore, they demonstrated that the more excitement and engagement the students felt during a presentation (TED videos), the more engagement was perceived.

Although our work also takes place in a learning environment, a virtual seminar held at multiple universities simultaneously, it differs from the studies which analyze the relationship between emotions and the learning process in that we investigate the relationship between the perceived quality of a presentation and the emotions among the audiences and the presenter. In other words, we focus on finding an emotional pattern that leads to a great presentation.

## 3. Data and Methods

### 3.1. Experimental Setup

We collected data from a virtual seminar called Collaborative Innovation Networks (COINs 2020) [43], which involved three instructors and 35 students from MIT, the University of Cologne and the University of Bamberg. Students formed virtual teams with three to five participants from different locations, resulting in a total of eight teams, each of which investigated a given complex business topic independently. Afterwards, the seminar was organized as a virtual meeting by using Zoom (https://zoom.us/, accessed on 10 May 2021) which was held every two weeks from 6 April 2020 to 14 July 2020. In each meeting, each team gave a PowerPoint-supported presentation of 10–15 min in rotating order, reporting the current progress of the team project to the audiences, i.e., the other teams and instructors.

During the virtual meeting, both the speakers and audiences were asked to keep the built-in camera on their local devices active to ensure that the faces would clearly appear on the video. We then recorded each meeting using Zoom's built-in recording function. In order to capture all the participant's faces, we recorded the video in Zoom's Gallery View, which can display up to 49 thumbnails in a grid pattern on a single screen. The recorded videos are the main analytical material used in this work.

After each presentation, the audience was asked to rate the overall performance by using a pre-designed anonymized poll published in a Google Form. Specifically, individuals were asked to answer "How many points will you give this presentation" on a numeric rating scale between 1 and 5, with 5 being the maximum number of points for the given presentation. Moreover, we provided an additional option, "I am a speaker", in the poll to prevent subjective scoring by the speaker. After collecting the poll data, we denoted the collective score y for a presentation as the mean value from all audiences. We deem this as a reasonable method to align the poll score with the presentation and most importantly, to reduce the bias of an individual's subjective evaluation. These ground-truth collective scores are used as the dependent variable for further analysis.

### 3.2. Data Pre-Processing

We applied different pre-processing steps to clean up the recorded data and to arrange them in a suitable form for subsequent analysis. Each recording was first divided into eight sequences and each of them contained the presentation of one group. We then converted each video presentation into a sequence of images. More specifically, we extracted one image (frame) per second from the video using Moviepy (https://zulko.github.io/moviepy/, accessed on 10 May 2021). As each image contained a grid of up to 49 faces, we localized and extracted individual faces from each image using face-recognition (https://github.com/ageitgey/face_recognition, accessed on 10 May 2021), which is a python li-

brary for detecting faces in a given image. Moreover, we utilized the same library to divide the individual faces into three groups, namely: audience, presenter, and supervisor. The library automatically detects similar faces given an example. Hence, we manually selected sample images for supervisor and presenter, respectively, and stored them separately using face-recognition. Note that in some presentations, neither the presenter nor the supervisor could be identified due to poor lighting conditions or due to the person of interest not being recorded at all. We then discarded images with low quality and this resulted in a total of 41 presentations with the number of individual faces ranging from 3600 to 15,863 for each presentation. Finally, we converted each image to greyscale and reshaped its size to 48 × 48 pixels. During model training, we used data augmentation, that is, randomly flipping the images horizontally (see Figure 1 for an illustration).

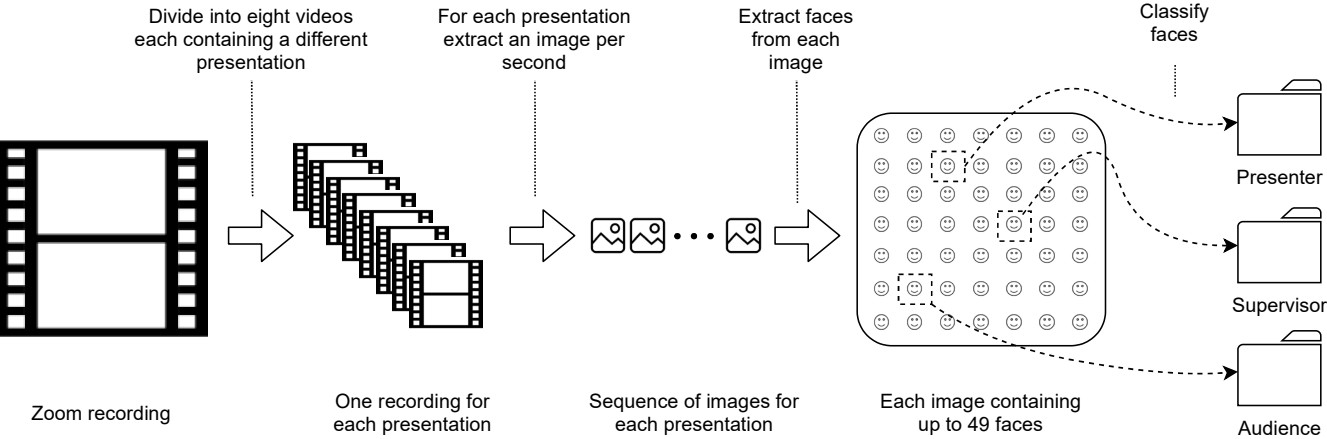

**Figure 1.** Overview of the data pre-processing steps. Each recording is divided into eight videos and each contains a different presentation. We then extracted one image per second for each presentation video. Finally, we extracted all faces from a single image and classified them into presenter, supervisor, and audience.

### 3.3. Facial Emotion Recognition

In order to identify participant's emotions, we used convolutional neural networks (CNNs) to employ facial emotion recognition (FER). In particular, we focused on six different emotions, inspired by Ekman and Oster [16], namely: anger, fear, happiness, neutral, sadness, and surprise. The classifier was trained on a variety of available FER data sets where each face was labelled manually with a corresponding emotion. Specifically, we collected the following data sets:

- 5405 images from AffectNet [19], which we labelled manually;
- 31,051 images from FERPlus (https://github.com/microsoft/FERPlus, accessed on 10 May 2021);
- 250 images from the Extended Cohn-Kanade Data set (CK+) [12];
- 184 images from the Japanese Female Facial Expressions (JAFFE) database [13];
- 522 images from BU-3DFE [14];
- 3455 images from FFQH (https://github.com/NVlabs/ffhq-dataset, accessed on 10 May 2021), which we labelled manually.

The images from AffectNet and FFQH were labelled by three researchers. All researchers received the same images and had to choose one of the following classes for each image: anger, fear, surprise, sadness, neutral, happiness, or unknown. The only images that were considered are those where two of the three researchers agreed on the same emotional state and ignoring images where the majority vote was on the unknown class.

Finally, we combined all images, which resulted in a large and heterogeneous FER data set containing 40,867 images (with 8.58% of anger, 3.26% of fear, 13.81% of surprise, 11.45% of sadness, 33.13% of neutral, and 29.77% happiness). Table 1 presents the distribution of emotional states along the six data sets. Note that we used 80% of the images for

training, 10% for testing, and 10% for validation. The validation set was used to estimate the prediction error for model selection. That is, the training of the model was terminated prematurely (before the maximum epoch) once the performance on the validation set became worse.

**Table 1.** Distribution of different emotions along the various FER data sets.

|  | Anger | Fear | Surprise | Sadness | Neutral | Happiness | Total |
|---|---|---|---|---|---|---|---|
| AffectNet | 473 | 512 | 1379 | 569 | 1873 | 599 | 5405 |
| FERPlus | 2606 | 648 | 3950 | 3770 | 11,011 | 9066 | 31,051 |
| CK+ | 45 | 25 | 83 | 28 | 0 | 69 | 250 |
| JAFFE | 30 | 32 | 30 | 31 | 30 | 31 | 184 |
| BU-3DFE | 92 | 92 | 89 | 88 | 84 | 77 | 522 |
| FFQH | 260 | 22 | 114 | 193 | 540 | 2326 | 3455 |
| Total | 3506 | 1331 | 5645 | 4679 | 13,538 | 12,168 | 40,867 |

Regarding the deep models, we considered several widely-used convolutional neural network (CNN) architectures including VGG [11] and Xception [44]. We created these models from scratch by utilizing Keras. We adopted the cross-entropy criterion as the loss function which is minimized using the Adam optimizer with a learning rate of 0.025. We trained each model up to 100 epochs. After training, we evaluated each model on the same testing set of the heterogeneous FER data set and chose the model with the highest performance in terms of accuracy. Subsequently, the selected model was used to predict the emotions for all faces that we had previously recorded in the 41 presentations.

### 3.4. Feature Engineering

In total, we calculated 18 audience and 6 speaker features for each presentation using the predicted emotions from the recordings. The audience features were created as follows. Firstly, for each emotion, we determined the ratio between the frequency of occurrences of a given emotion by the audience and all the emotions the audience expressed during the presentation. We denote this feature as ratio_audience (E), where E refers to a specific emotion, such as anger, fear, surprise, sadness, neutral, or happiness. For each emotion expressed by the audience, we then calculated the number of times it occurred at least once per second in a given presentation. Subsequently, we put this number in relation to the total recorded time of the same presentation. We denote this audience feature as density (E), with E referring to an emotion. Finally, for each presentation, we normalized the frequency of an emotion expressed by the audience between 0 and 1 and calculated its standard deviation over the entire presentation. This feature is referred to as deviation (E), with E representing the given emotion.

Since there is only one speaker at a given second in a presentation, we could not compute the same features for the speaker as we could for the audience. More specifically, we only calculated the ratio between the frequency of a specific emotion the speaker expressed and the number of all recorded emotional states the speaker exploited during the presentation. We denote this feature as ratio_speaker (E), where E refers to a given emotion.

The above described features were used to calculate correlations between audience, speakers, and presentation scores. We also used some of the features for an ordinary least squared regression.

### 4. Results

We found that the VGG16 model performed best with a test accuracy of 84.0%, closely followed by the Xception model with 83.7%, the VGG19 model with 83.6%, and the VGG13 model with 82.6% (see the confusion matrices on the test set for each model in Figures 2–5). Figure 2 illustrates the confusion matrix on the test set for the VGG16 model including

the in-class precision for each emotion: 84% neutral, 93% happiness, 83% surprise, 70% sadness, 83% anger, and 64% fear.

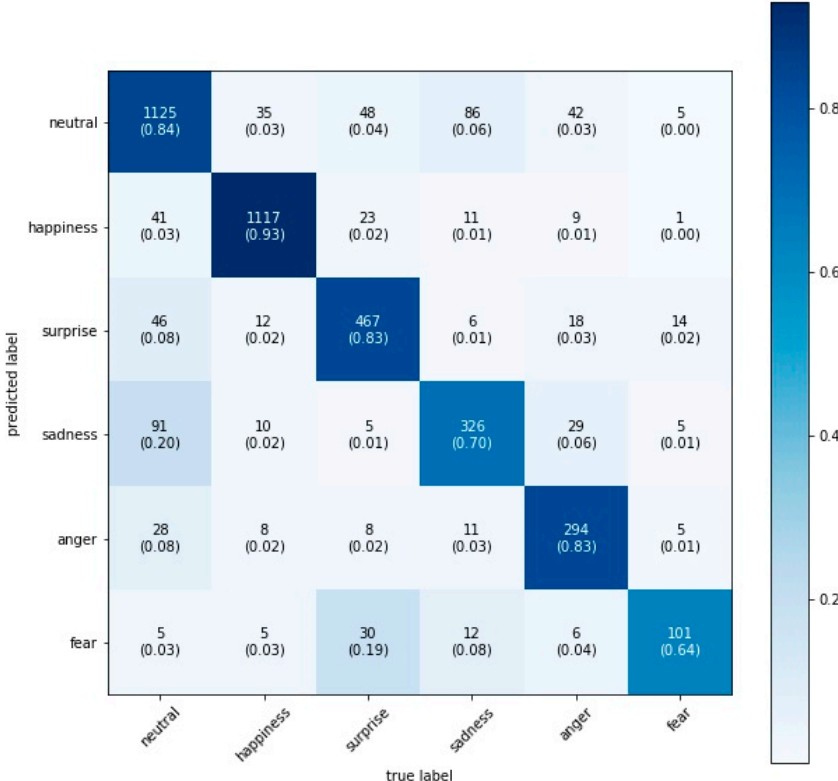

**Figure 2.** VGGs16 confusion matrix on the test set.

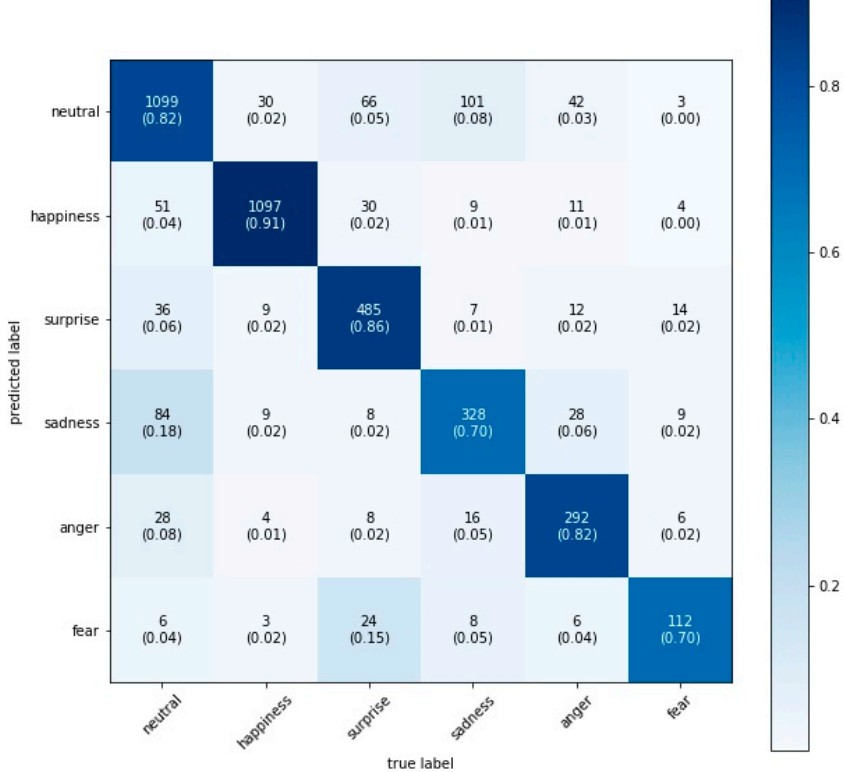

**Figure 3.** VGGs19 confusion matrix on the test set.

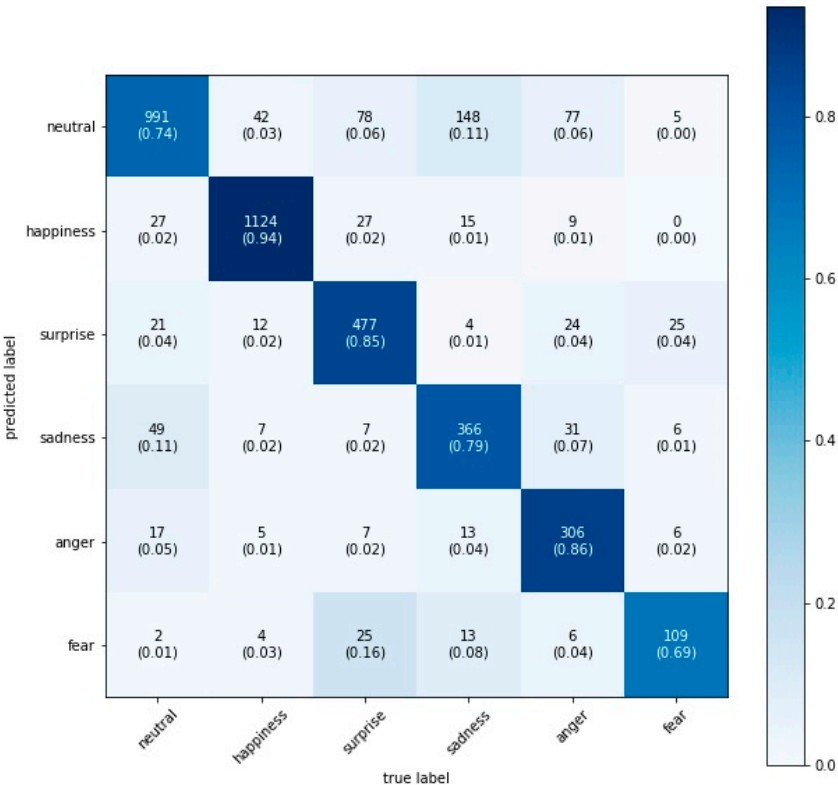

**Figure 4.** VGGs13 confusion matrix on the test set.

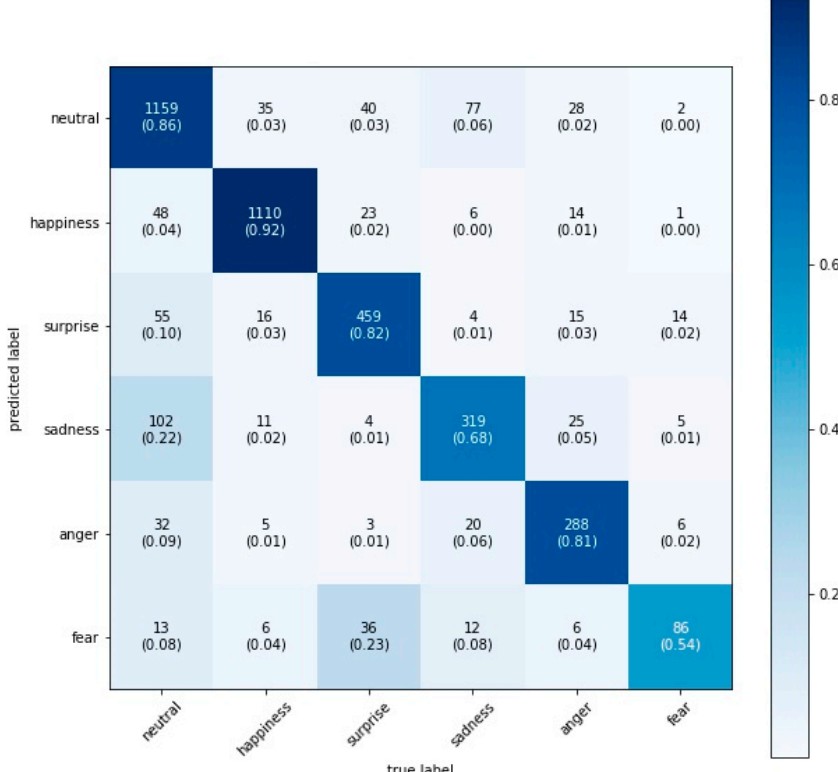

**Figure 5.** Xception confusion matrix on the test set.

After predicting the emotions of all faces recorded during the Zoom presentations using the VGG16 model, we calculated the 18 audience features and the 6 speaker features described in Section 3.4. We first correlated the emotions of the audience with the emotions

of the speakers for each presentation using the ratio_audience(E) and ratio_speaker(E) features. The coefficients are captured in Table 2. We found that, firstly, the happier the speaker was, the happier and less neutral the audience was and, secondly, the more neutral the speaker was, the less surprised the audience was.

**Table 2.** Correlation coefficients between the emotions of the audience and the emotions of the speaker along all presentations (N = 41) (*** Correlation significant at 0.01 level; ** Correlation significant at 0.05 level).

|  | Ratio_Speaker (Happy) | Ratio_Speaker (Neutral) | Ratio_Speaker (Fear) | Ratio_Speaker (Sad) | Ratio_Speaker (Surprise) | Ratio_Speaker (Angry) |
|---|---|---|---|---|---|---|
| ratio_audience(happy) | 0.5107 *** | −0.0754 | 0.0544 | −0.0532 | −0.2369 | −0.2082 |
| ratio_audience(neutral) | −0.4347 *** | 0.0909 | −0.0432 | −0.0283 | 0.1850 | 0.1340 |
| ratio_audience(fear) | 0.2240 | 0.0983 | −0.1953 | 0.0210 | −0.3097 | 0.1471 |
| ratio_audience(sad) | −0.1185 | 0.0592 | 0.0165 | −0.0780 | −0.0153 | 0.2119 |
| ratio_audience(surprise) | 0.2059 | −0.3147 ** | 0.1163 | 0.1995 | 0.1952 | 0.1602 |
| ratio_audience(angry) | −0.0123 | 0.0013 | −0.0569 | 0.1650 | 0.0089 | −0.1450 |

Next, we computed the correlations between all features (18 audience and 8 speaker features) and the presentation scores. The significant features, their coefficients, and *p*-values are provided in Table 3.

**Table 3.** Correlation coefficients between the emotions of the audience and speakers and the presentation scores along all presentations (N = 41).

|  | Presentation Score | *p*-Value |
|---|---|---|
| deviation (happy) | 0.73 | $7 \times 10^{-8}$ |
| deviation (neutral) | 0.50 | $8 \times 10^{-4}$ |
| deviation (fear) | 0.34 | 0.025 |
| ratio_audience (happy) | 0.55 | $2 \times 10^{-4}$ |
| ratio_audience (neutral) | −0.44 | $4 \times 10^{-3}$ |
| ratio_audience (fear) | 0.38 | 0.010 |
| density (happy) | 0.44 | $4 \times 10^{-3}$ |
| ratio_speaker (happy) | 0.35 | 0.026 |

We found that the deviation in the audience's happiness, neutrality, and fear; the ratio in the audience's happiness and fear; the density in the audience's happiness as well as the ratio in speaker's happiness are positively related with the presentation score. Contrarily, the ratio in the audience's neutrality is negatively correlated with the presentation score. All of these correlations are significant with a *p*-value smaller than 0.05. The correlation between the deviation in audience's happiness and the presentation score is the most significant with a coefficient of 0.73 and a *p*-value of $7 \times 10^{-8}$. Recall that the deviation in audience's happiness measures the variation in happiness throughout the presentation. The latter correlation is also illustrated in Figure 6, where we plotted the development in happiness for some selected presentations. We can see that the more variation in the audience's happiness we have, the better the score (see the blue dot in the upper right corner of Figure 6a) and vice versa (see the blue dot in lower left corner of Figure 6a).

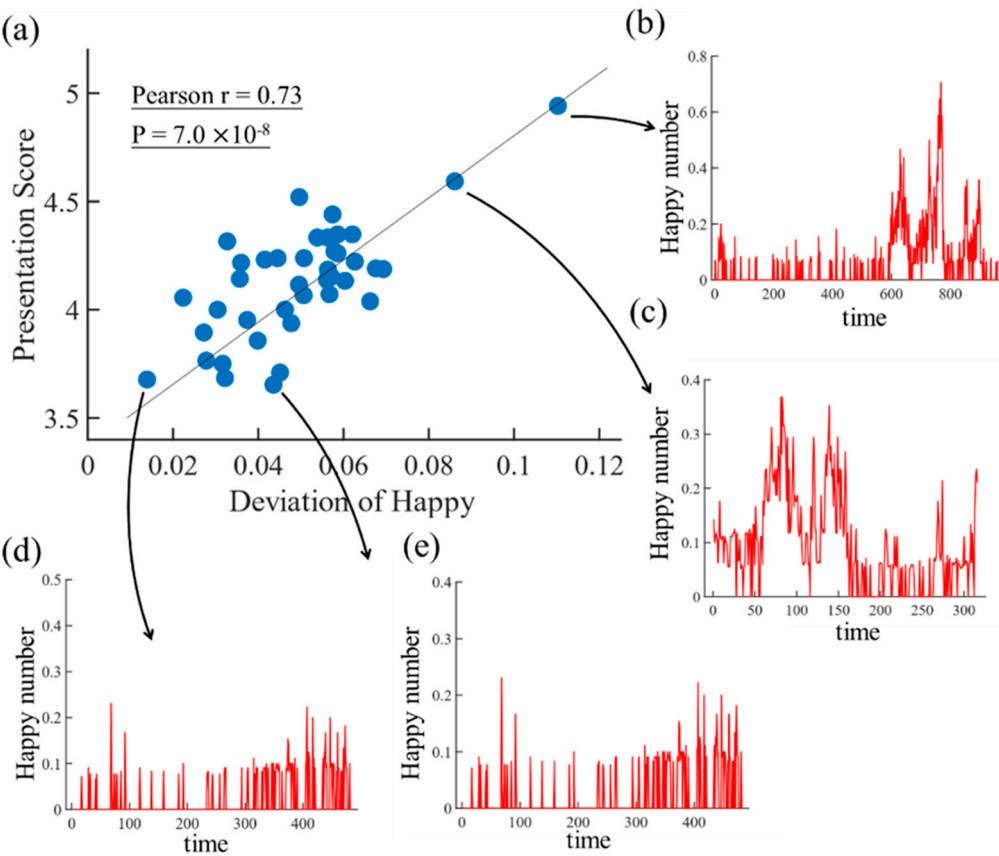

**Figure 6.** (**a**) Illustration of the correlation between presentation score and the audience's deviation in happiness. We can see that presentations which caused wide fluctuation in happiness among the spectators achieved higher scores (**b**,**c**) than presentations which did not cause fluctuation in happiness among the spectators (**d**,**e**).

Finally, we used an ordinary least square (OLS) regression to predict the presentation score using the audience's deviation in happiness and the audience's deviation in fear as independent variables. Both features were chosen after several tests. A summary of the OLS regression model is provided in Table 4. We found that the R2 of the OLS model was 0.55 which indicates that the regression model described 55% of the variation in the presentation scores using only two features, namely the deviation in the audience's happiness and fear.

**Table 4.** Summary statistics of the OLS regression with the audience's deviation in happiness and the audience's deviation in fear as independent variables and the presentation score as the dependent variable (N = 41).

| Variables | Coefficient | Standard Error | T-Statistics | *p*-Value |
|---|---|---|---|---|
| Intercept | −0.076 | 0.071 | −1.063 | 0.295 |
| deviation (happy) | 7.866 | 1.411 | 5.574 | $2.8 \times 10^{-6}$ |
| deviation (fear) | 12.026 | 6.372 | 1.888 | 0.067 |

## 5. Discussion

In this work, we have experimentally verified our hypothesis that certain combinations of emotions conveyed by the presenter, or shown by the audience, are indicators of higher perceived presentation quality. Based on recent advances in FER in deep learning based methods, particularly CNN, we compared the emotions of participants in a seminar taught over a semester through their facial expression captured in videoconferencing. As the regression coefficients in Table 4 show, the larger the deviations of happiness and fear among the audience, the higher the is presentation scored. In other words, if the spectators

experience a broad range of emotions with wide fluctuations in happiness and fear, they scored the presentation the highest. The same insight can also be gained from Table 3, which correlates the presentation score with the different emotions. We find that neutral faces, probably indicating the boredom of the audience, are negatively correlated with the score. The highest positive correlation with the presentation score is, again, found with the deviation of happy, neutral, and fear, which confirms the regression result. Surprisingly, the appearance of fear on the faces of the audience correlates positively with the perceived quality of the presentation. Simply providing constant happiness is also positively associated with the average score of all presentations and this is shown by the positive correlation of ratio_audience (happy) with the score. Presenters achieve the most engaged audience and obtain the highest score when they smile—showing a happy face, illustrated by the positive correlation between the happiness in the face of the speaker and the score of its presentation. Happy presenters will also reduce neutral faces among the audience, as illustrated in Table 2. Having fewer neutral faces is associated with a higher presentation score.

## 6. Limitations

While this project provides interesting insights, much further work is needed. In particular, we are not making a claim with respect to causality. Rather, we take triggering a wider range of emotional expressions of the audience as an indicator of a successful presentation. In order to solidly claim a causal link between triggering a broad range of emotions to produce a great presentation, we would need to conduct controlled experiments, for instance comparing presentations triggering fear by showing horror movie snippets with presentations including humor parts. Nevertheless, there are results from other areas, such as marketing and advertisement, where it has been found that emotional advertisements are more successful [45], suggesting that there might indeed be a causal link between experienced emotionality of the audience and presentation quality.

An additional limitation is the low number of measurements which number to 41. In an ideal world, the experiments should be repeated with a larger N to gain more statistical significance. Furthermore, our training data set for the CNN was unbalanced as there were ten times more happy and neutral faces in the dataset than fearful faces. However, as shown in the confusion matrices in the results section, this did not negatively impact our accuracy. Additionally, it is well known that facial recognition systems, which are not restricted to emotion recognition, are biased towards Caucasian males [46] and discriminate against females and non-Caucasians. Finally, it is also worth mentioning that video presentations are restricted in communicating emotions since there are only the contents of the PowerPoint slides and the voice and face of the presenter in the little "talking head" to get emotionality across to the audience. In real-classroom scenarios, the body language of the presenter and other interaction channels and environmental cues, such as smell, convey a much richer emotional experience for the audience; this introduces other factors that influence the perceived quality of a great presentation.

## 7. Conclusions

This work contributes to alleviating the restrictions imposed by COVID-19, by trying to develop recommendations to tackle "videoconferencing fatigue" resulting from long video meetings because of home office work. While there is no substitute for face-to-face interaction, we have tried to identify predictors of more highly rated and, thus, less stressful video meetings. We find that trying to provide "unlimited bliss" by keeping the audience constantly happy is not the best method for high-quality video presentations. Rather, a good presenter needs to challenge the audience by puzzling it and providing unexpected and even temporarily painful information, which then will be resolved over the course of the presentation. On the other hand, the presenter should constantly provide a positive attitude to convey enthusiasm and positive energy to the audience. In this manner, even

lengthy video meetings will lead to positive experiences for the audience and, thus, results in a similar experience for the presenter.

**Author Contributions:** P.G. conceived of the project; J.S. and J.R. designed the experiments and analyzed the results; P.G., J.S. and J.R. wrote the manuscript. All authors have read and agreed to the published version of the manuscript.

**Funding:** This research has received no external funding.

**Institutional Review Board Statement:** The study was conducted according to the guidelines of the Declaration of Helsinki and approved by the Institutional Review Board of MIT (protocol code 170181783) on 27 March 2019.

**Informed Consent Statement:** Informed consent was obtained from all subjects involved in the study.

**Data Availability Statement:** The data presented in this study are available upon request from the corresponding author. The data are not publicly available due to privacy reasons.

**Acknowledgments:** We thank Yucong Lin for his assistance in collecting the data. We also thank the students in the course for participating in our experiment.

**Conflicts of Interest:** The authors declare no conflict of interest.

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
