# Peer review of "Reducing Videoconferencing Fatigue through Facial Emotion Recognition"

_futureinternet, doi:10.3390/fi13050126_

Round 1

Reviewer 1 Report

The manuscript offers an interesting work about emotion recognition in Zoom meetings. The state of the art lacks some basic concepts and references, easily fixable. The introduction is sufficient to introduce the techniques, including the references introduced in subpars. 2.1 and 2.2 (which content can be eventually moved and reframed): focusing the analysis on specific works looks incomplete, especially in face recognition, where emotion recognition lists many works. A better literature analysis will give more focus on the different approaches (for instance dividing them in deep learning and other classification methods) instead of specific works, giving reference to 2-5 works for each approach.

The reference list is geographically biased, most but not only on Asian works, thus some European ones are suggested to balance it: Italian in this case because Italy is one among the first European nations to have initiated research in Emotion Recognition. Please consider them as suggestions for better reading and analysis, not as must-include papers, but consider the requirement to balance the works list provenience. Good works also come from Australia.

The value of the work is sufficient but there is a major issue in conclusions, which are not supported by data. Please keep in mind for any future work that correlation is not causation. Please investigate this statistical concept further with a web search.

1) please capitalize Zoom in the paper, otherwise, it's meant to be the magnifying glass tool).

2) The experimental work is done using the Ekman model (please cite and add a reference to the Ekman model, which is the model of the six emotions that you use)

3) on a sufficient number of participants (40 is recommended, 36 is in the paper), on 8 different groups in a course, thus with a cultural bias (please add it in a limits paragraph). 

4) For the state of the art about emotion recognition in political leaders (from which you start your paper), see:

Francesca D'ErricoIsabella Poggi:
Tracking a leader's humility and its emotions from body, face and voice. Web Intell. 17(1): 63-74 (2019)

OR

Francesca D'ErricoOliver NiebuhrIsabella Poggi:
Humble Voices in Political Communication: A Speech Analysis Across Two Cultures. ICCSA (2) 2019: 361-374

5) "Emotions in presentations can be expressed through multiple ways: The choice of words, e.g. positive vs. negative words"...

There should be a clear distinction between Emotion Recognition (of which you are speaking in the paper) and Sentiment Analysis (positive-negative-neutral, which is not the same context of recognition).

For this distinction, see:

Jordi VallverdúAlfredo Milani:
Errors, Biases and Overconfidence in Artificial Emotional Modeling. WI (Companion) 2019: 86-90

For the state of the art of Emotion Recognition in words, see:

Giulio BiondiAlfredo Milani:
A Web-Based System for Emotion Vector Extraction. ICCSA (3) 2017: 653-668

Giulio Biondi, Valentina Poggioni:
A Deep Learning Semantic Approach to Emotion Recognition Using the IBM Watson Bluemix Alchemy Language. ICCSA (3) 2017: 718-729

Yuanxi LiPaolo MengoniAlfredo Milani:
Clustering Facebook for Biased Context Extraction. ICCSA (1) 2017: 717-729

6) " facial expressions the presenter makes with his body and face, e.g. looking happy vs. sad"

Emotion Recognition is a broad and now well-studied field among the novel ones, going much deeper than a distinction between only two classes at a time (two classes can be randomly guessed with a 50% hit rate: usually even trivial sentiment analysis uses at least a third class, neutral, to balance the results, while Emotion Recognition very often classifies all the Ekman model or its subsets greater than two elements). "Happy Vs. sad" does not seem a proper example.

For face-based emotion recognition on the Ekman model, see for instance the following papers and their reference lists:

Giulio BiondiDamiano PerriOsvaldo Gervasi:
Enhancing Mouth-Based Emotion Recognition Using Transfer Learning. Sensors 20(18): 5222 (2020)

Osvaldo Gervasi, Matteo RiganelliSergio Tasso:
Automating facial emotion recognition. Web Intell. 17(1): 17-27 (2019)

7) "While most of the literature analyses the former, research on the relationship between speaker and listeners facial expressions and the quality of the presentation appears to be scarce. "

There is broad literature about it in e-learning. Please see for instance the following papers and their reference lists:

Berardina De CarolisFrancesca D'ErricoNicola MacchiaruloMarinella PacielloGiuseppe Palestra:
Recognizing Cognitive Emotions in E-Learning Environment. HELMeTO 2020: 17-27

Berardina De CarolisFrancesca D'ErricoMarinella PacielloGiuseppe Palestra:
Cognitive Emotions Recognition in e-Learning: Exploring the Role of Age Differences and Personality Traits. MIS4TEL 2019: 97-104

Berardina De CarolisFrancesca D'ErricoNicola MacchiaruloGiuseppe Palestra:
"Engaged Faces": Measuring and Monitoring Student Engagement from Face and Gaze Behavior. WI (Companion) 2019: 80-85

as well as the already cited political presentations. 

8) The focus of the work should be better cleared between "marketing" (an understatement to make more impressive presentations) or investigation of the relationship between emotions and the quality of presentations (which should be stated in the limits paragraph as definite only for remote ones, which is not applicable to presentations in presence, which takes more advantage of non-verbal communication).

8) The experimental setup seems accurate and proper for the research topic, giving objective classification with a knowledge base on subjective evaluations via questionnaire. Objective and subjective results may differ (this can lead to some margin of classification error) and the facial expression during the presentations can vary on different stimuli but the approach seems adequate.

9) Please give a motivation on the scale choice: "on a scale between 1 and 5".

Information about statistical scales for Emotion Recognition can be found here: 

Alfredo MilaniGiulio Biondi:
SEMO: a semantic model for emotion recognition in web objects. WI 2017: 953-958

10) This labelling work:

"5,405 images from AffectNet [23] which we labelled manually by ourselves"

"3455 images from FFQH5 which we labelled manually by ourselves"

could be a great job or awful work. For sure the effort is great and should be shared: if possible, please share a link to the labelled dataset. To state if the results are as good as the effort, the labelling procedure should be explained in details: if it's adequate, it can be underlined as one of the values of this work, otherwise, it should be listed in the limits paragraph. In fact, if not properly conducted, it could have introduced additional bias.

11) About the other image collections, biases about provenience, gender or "fake emotions" (which means performed by actors, see: 

Jordi VallverdúAlfredo Milani:
Errors, Biases and Overconfidence in Artificial Emotional Modeling. WI (Companion) 2019: 86-90

) should be listed in the limits paragraph. The collecting effort could have reduced eventual biases (if so, the original bias and the reduced bias from mixing collections should be underlined), but generally speaking, "fake emotions" is a bias present in most of the available datasets. Another limit is the unbalance of the final classes, which is not a problem for CNN-based deep learning, but should be underlined for the sake of future researches which want to compare with your work.

12) It's not clear how you distinguish test and validation cases, and how you use in the classification process the subjective evaluation from the quizzes.

13) "That is, the input size of each model is set to 48*48 and the output layer’s size is set to 6. 233
All models are trained from scratch using data augmentation, which is based on including horizontally flipped images. "

Layers size and data augmentation can be moved to the preprocessing paragraph (being antecedent to the processing phase of the classification, where images are given as input to the CNN).

14) Among evaluation metrics, precision and accuracy (and F-1) are the most relevant because the most used (thus, useful for comparison with other works), while the emotion recall is scarcely significant in this case.

15) "we plotted the development in happiness for some selected presentations. We can see that, the more variation we have, the better the score"

Please clarify variations in what.

16) "Surprisingly, the appearance of fear on the faces of the audience correlates positively with the perceived quality of the presentation. This means that the audience needs to be shocked occasionally by the presenter. "

Here I see one of the most common but most serious errors in data interpretation: you interpret correlation as causation. Correlation is not causation, and even less you can conclude that "the audience needs to be shocked". This is a major naïve mistake that should be totally avoided.

The motivation of the correlation can be casual or due to other reasons (for instance, misclassification of the "fear" class, which often happens from neutral expressions, see:

Giulio BiondiDamiano PerriOsvaldo Gervasi:
Enhancing Mouth-Based Emotion Recognition Using Transfer Learning. Sensors 20(18): 5222 (2020)

together with all the work of Paul Ekman).

17) "This means, unsurprisingly, that while providing an emotional rollercoaster is best, it is certainly better to make the audience happy than sad. "

The first part of the sentence presents again the "correlation is causation" major mistake, while the second part seems too trivial to be noted.

Since the paper title is based on the word "roller-coaster", the title should be changed. The word is neither scientific nor does convey a meaning supported by the data.

18) All the related content in the conclusion should be removed: "We find that trying to provide “unlimited bliss” by keeping the audience constantly happy it not the best way for high-quality video presentations. Rather a good presenter needs to challenge the audience, by puzzling it and providing unexpected and even temporarily painful information, that then will be resolved over the course of the presentation. On the other hand, the presenter should constantly provide a positive attitude, to convey enthusiasm and positive energy to the audience. This way, even lengthy video meetings will lead to positive experiences for the audience, and thus also for the presenter" 

not being supported by data but guessed conclusions.

19) The part adding the social network starting from line 341 seems totally out of topic and adds confusion to the study. I suggest totally eliminate it, which does not seem to give proper added values, nor being a sufficiently broad and connected network to lead to proper prediction. The connected self-references should also be avoided, being out of topic for this paper.

If authors do not want to avoid this further analysis, this should be stated in all the paragraphs, introducing this step in the introduction, explaining it deeply in settings and adding results to the discussion with a good separation of the steps and of results, and finally adding the fact that the network is small in the limits paragraph.
Probably the best effort should be to eliminate it, to avoid wrongly looking only as a way to self-cite.

Author Response

see attached response letter

Reviewer 2 Report

Dear authors,

Please find the attached file for my comments. Please update the paper based on the comments and resubmit it.

Best Regards 

Author Response

see attached response letter

Reviewer 3 Report

The manuscript "Reducing zoom fatigue through an emotional roller-coaster using facial emotion recognition" is well-written and clear. The topic is original and pertinent because of the sanitary context but contribute for research on emotions more generally. I have some comments :

  • in the title, the reference to Zoom do not seem adequate even if it is the material used ; it is restrictive. I propose do replace "Zoom"  by "virtual meeting"
  • the authors do not propose hypotheses
  • in the discussion, the authors should recall the objective of the study

Author Response

see attached response letter

Round 2

Reviewer 1 Report

I thank the authors for the detailed and clear cover letter.

The review requests have been addressed. Only a few claims still remain confused in the paper:

1) "This means that the audience needs to be shocked occasionally by the presenter." As previously noted, this claim is a personal deduction of authors which is not adequately supported by data. The following sentence "It might also be related to".... is not sufficient to moderate the "it means that .... needs...." inadequate claim which should be removed because it was not reframed.

2) "This means, unsurprisingly, that while providing an emotionally
diverse experience is best, it is certainly better to make the audience happy than sad." This sentence is trivial and not significant for discussion and conclusions, and should be removed. 

3) Limitations should be moved as previously asked in a separate paragraph, due to their importance.

4) The conclusion "Nevertheless, we think that these restrictions actually assist in focusing on key determinants of successful presenters. Based on these emotional insights gained from facial emotion recognition, we find that attaining more emotionality among the audience indicates better presentations, confirming the saying “no pain, no gain”" does not look scientific and should be removed, lacking soundness and lowering the scientific style and presentation level.

Author Response

please see attached response letter

Reviewer 2 Report

Dear Authors,

Thank you for addressing all my comments and the current form of the paper is accepted for publication.

Best Regards 

Author Response

please see attached response letter
